# Microplastics: A Threat for Male Fertility

**DOI:** 10.3390/ijerph18052392

**Published:** 2021-03-01

**Authors:** Stefania D’Angelo, Rosaria Meccariello

**Affiliations:** Department of Movement Sciences and Wellbeing, University of Naples Parthenope, 80133 Naples, Italy; stefania.dangelo@uniparthenope.it

**Keywords:** microplastics, endocrine disrupting chemicals, reproduction, spermatogenesis, spermatozoa

## Abstract

Much of the planet is swimming in discarded plastic, which is harming animal and possibly human health. Once at sea, sunlight, wind, and wave action break down plastic waste into small particles: the microplastics (MPs). Currently, particular attention has been drawn to their effects on aquatic environments but the health risks, especially in mammals, are poorly known. These non-biodegradable materials can act as a vector for environmental pollutants, can be ingested by humans in food and water, and can enter and accumulate in human tissues with a possible risk for heath. Recent studies revealed the deleterious effects of MPs exposure in male reproduction and sperm quality, making them a potential hazard to reproductive success. This manuscript summarizes the main changes in sperm quality along the lifespan and the upcoming studies on the effects of MPs in male fertility in mammals.

## 1. Introduction

One of the main consequences of industrialization is the production, use and discharge of several environmental pollutants that can result as harmful for animal, human and environmental health.

Many environmental pollutants can act as endocrine disrupting chemicals (EDCs), mimicking the activity of endogenous steroid hormones and interfering in the endocrine functions with different mechanisms [1,2,3,4,5,6,7,8,9]. In recent years, particular emphasis was placed on plasticizers, plastic additives, and contaminants of emerging concern (CECs) which includes pharmaceuticals, personal care products, food additives, natural and synthetic hormone and plastic debris in micro and nano range, among the others that are directly or indirectly discharged into to the environment [10]. In this respect, chemicals like phthalates, bisphenols poly- and perfluorinated alkyl substances, among others, are commonly used for the production of daily use goods and are therefore frequently released into the environment as waste. For example, bisphenol A (BPA), a plasticizer used for the synthesis of phenol resins, polyacrylates, polyesters, epoxy resins, and polycarbonate plastics, is used for the production of drink and food packaging, and in case of high temperature exposure or pH variation (e.g., washing in washing machines, food heating into the microwave, contact with acid foods) leaches into wastewater, contaminates foods and beverages, thus representing both ecotoxicological and health risks [4]. Oxidative stress-induced tissue damage and consequent apoptosis, poor gamete quality, developmental abnormalities, neurotoxicity, metabolic disorders or epigenetic changes are some of the direct and in utero exposure effects of EDCs [1,2,3,4,5,6,7,8,9].

In recent years, the growing rate of infertility has displaced attention to gametogenesis and gamete quality. Once considered a “woman’s trouble”, infertility is on the rise in males and semen quality has declined in recent decades [11]. In about 40% of men with impaired spermatogenesis, the etiology remains unknown after a complete diagnostic work-up [12]. Spermatozoa (SPZ) are not only the carrier of a haploid nucleus into egg cells, but emerging evidence has revealed that SPZ contribute to early embryo development and offspring health with an epigenetic signature highly sensitiveto both environmental factors, including EDCs, and paternal lifestyle [4]. Decreased sperm quality has been reported in subjects exposed to environmental pollutants [13,14] (for review), suggesting that the evaluation of sperm quality may represent a biomarker for general health [13,14].

Currently, particular attention has been drawn to the effects of microplastics (MPs) on the aquatic environment; however, the health risks of these environmental pollutants—especially with respect to mammalian reproduction—are poorly known. However, in the past three years, preliminary assessments of the effects of MPs exposure in mammalian reproduction have emerged with the publication of peer-review articles that revealed the effects on spermatogenesis and sperm quality in exposed animal models and the indirect effects on the offspring occurring via gestational exposure. This manuscript summarizes the main ecotoxicological and health risk of MPs in mammals, the main threat for sperm quality along the lifespan and the upcoming studies on the effects of microplastics (MPs) in male fertility in mammals.

## 2. MPs: An Emerging Threat for Health

About 6300 million tons of plastic waste was generated between 1950 and 2015. Most of this waste, about 4900 million tons, ended up in landfills and the environment. Based on the trends of that period, the researchers estimated that by 2050 the amount of plastic waste in landfills and the environment would reach 12,000 million tons. Nonetheless, the potential dangers of escalating plastics pollution, particularly MPs pollution, have remained largely ignored by governments and policy makers [15,16].

Use of plastics in everyday items and manufacturing processes has resulted in a deluge of slowly degradable materials entering our environment and our food chain. As plastics breakdown into tiny particles the consequences on human, animal and ecosystem health need to be studied [17].

Plastics are synthetic organic polymers. Their long-term durability, increasing scale of production, unsustainable usage coupled with the inadequate waste management systems have led to the accumulation of plastics in ecosystems worldwide [18,19]. Plastic is the most prevalent type of marine debris found in our ocean and the Great Lakes. Plastic debris can come in all shapes and sizes. However, the most important and emerging threat posed by plastic pollution is the breaking down of plastic into smaller pieces called MPs.

MPs have been identified in many food substances like salt (50–280 microparticles/kg of salt), branded milk (6.5 ± 2.3 particles/L), fish and other seafood, and tea from teabags (11.6 × 109 s/plastic teabag) [20]. According to a recent study, three coffees in disposable paper cups are enough to make us ingest about 75 thousand microplastic particles. The commonly used paper cups have a thin layer of plastic which, in contact with the hot liquid, releases MPs [21]. The researchers poured boiling water into some shot glasses. After 15 min, they proceeded with analyzing the water based on the possible presence of MPs and additional ions. They observed that about 25,000 micron-sized microplastic particles are released into 100 mL of hot liquid (85 to 90 °C). Simply put, the average person who drinks three cups of coffee a day, in paper cups, ingests around 75,000 tiny microplastic particles [21]. MPs act as carriers of contaminants such as ions, toxic heavy metals, and hydrophobic organic compounds which, if ingested regularly, can have serious health implications. Dangerous MPs are not a specific kind of plastic, but rather any type of plastic fragment that is less than 5 mm in length according to the European Chemicals Agency and the U.S. National Oceanic and Atmospheric Administration (NOAA); they enter natural ecosystems from a variety of sources, including cosmetics, clothing, and industrial processes [22]. MPs pollution is the one of the most challenging ecological threats the next generation will face. MPs are divided into two types: primary and secondary. Examples of primary MPs include microbeads found in personal care products, plastic pellets (or nurdles) used in industrial manufacturing, and plastic fibers used in synthetic textiles (e.g., nylon). Primary MPs enter the environment directly through any various channels—for example, product use (e.g., personal care products being washed into wastewater systems from households), unintentional loss from spills during manufacturing or transport, or abrasion during washing (e.g., laundering of clothing made with synthetic textiles). Secondary MPs form from the breakdown of larger plastics; this typically happens when larger plastics undergo weathering, through exposure to, for example, wave action, wind abrasion, and ultraviolet radiation from sunlight. MPs are not biodegradable. Thus, once in the environment, primary and secondary MPs accumulate and persist.

Apart from aquatic ecosystems that are directly and daily exposed to plastic debris with different outcomes on the health of aquatic microorganisms, plants and animals, the exposure risk for terrestrial species also emerged [23,24].

Recently, MPs have been observed in atmospheric fallout and also in drinking water as well as in drinking water sources [25,26]. In particular, fibers are the dominant shape of MPs in the atmosphere and synthetic textiles are the main source of airborne MPs which widely distribute in the environment in consequence of atmospheric conditions and human activities. In addition, airborne MPs further contribute to the pollution of the aquatic environment [25]. As a consequence, the exposure routes have been expanded from the food-chain to contaminated food and drinks and inhalation. Emerging evidence revealed the presence of MPs in human stools [27] and colectomy specimens [28], confirming that human exposure to MPs through ingestion does at least occur. Figure 1 summarizes the main exposure route of MPs for human.

Tissue-accumulation kinetics and distribution pattern strongly depend on the size ofMP particles. Therefore, studies on MPs have been recently translated into mammals and cell lines, revealing toxicity effects in different cell lines and the main effect of gut–microbiota (i.e., dysbacteriosis and inflammation), liver, and kidney in animal models [23], for recent review], with exposure-induced oxidative stress, inflammation and interference in energy and lipid metabolism, altered blood biomarkers and neurotoxicity [23,29,30]. Ovarian fibrosis, apoptosis and pyroptosis of granulosa cells are the main consequences of MPs-induced oxidative stress in female rats [31,32].

Interestingly, MPs exposure in parent mice and dams caused tissue damage and disturbed immune response; reduced number of live births, altered sex ratio, decreased body weight and changes in lymphocyte composition within the spleenwere observed in the offspring [33]. Similarly, maternal MPs exposure during pregnancy and lactation in rodents resulted in liver damage, metabolic disorders, gut barrier dysfunctions and microbiota disorders in the exposed dams and metabolic disorders in the F1 and F2 generations [34], thus revealing long-term intergenerational effects occurring via placenta or maternal milk exposure. The first report of MPs in human placenta has recently been published [35]. Although this study is limited to six samples only, 12 microplastic fragments (5–10 μm in size), were found in four placentas, in chorioamniotic membranes, maternal side and fetal side, suggesting possible consequences on gestation, embryo developmentand health [35].

In this respect, MPs represent an upcoming ecotoxicological trouble and a health risk; their impact deserves attention to preserve reproductive health, pregnancy and the disease load of exposed subjects and offspring. In the next paragraph, we focus on the preservation of male fertility along lifespan and analyze upcoming reports concerning the effects of MPs on mammalian male reproduction.

## 3. Sperm Quality along Lifespan

Successful reproduction depends on the production of high-quality gametes. In human, the production of SPZ occurs throughout life starting from puberty and requires an orderly succession of mitotic proliferation, meiotic division and differentiation events within the testis and a further phase in male and female reproductive tract to make the SPZ acquire motility and fertilizing ability. Intricate endocrine, paracrine and autocrine signaling networks are responsible for the production of high quality SPZ and the process is susceptible to the modulation by lifestyle and environmental factors [4,5,6,36,37,38,39,40,41,42].

Oxidative stress together with the age-dependent decrease in antioxidant activity and mitochondria dysfunctions are the main causes of testicular and sperm damage. In fact, reactive oxygen species (ROS) overproduction is responsible for spermatogenesis failure, the apoptotic loss of both germ and somatic cells, oxidative DNA damage, failure in gene expression and post-transcriptional gene regulation, or APT depletion. As a consequence, the functional impairment of SPZ occurs, with SPZ exhibiting insufficient axonemal phosphorylation in sperm tail, lipid peroxidation, loss of sperm motility and viability, among others [43,44,45,46]. Hydrogen peroxide, but also superoxide anion, are the main ROS detected in the sperm of infertile patients [47]. However, SPZ require a high energy level and controlled ROS are physiologically involved in sperm maturation in male and female reproductive tracts; processes like epididymal transport, SPZ maturation, capacitation, acrosome reaction, and signaling processes to ensure fertilization all require ROS activity [48]. Nevertheless, the lacking capacity of DNA repair in sperm and the high content in polyunsaturated fatty acids in the membrane make spermatozoa highly sensitive to oxidative stress damage [46]. However, the seminal fluid contains both enzymatic and non-enzymatic antioxidant defenses like vitamins A, C and E, superoxide dismutase, catalase, or glutathione peroxidase and reductase [46]; in parallel, molecular chaperones and cochaperones, ubiquitinating and deubiquitinating enzymes deeply contribute to the preservation of spermatozoa quality [49,50].

Lifestyle and environmental factors like smoking, alcohol consumption, stress, diet composition, sedentary life, environmental pollutants, EDCs, heavy metals, or abuse of illicit substances, interfere with spermatogenesis leading to poor sperm quality and infertility with possible consequences on the offspring [3,4,5,6,42,45,51]. Interestingly, the potential to change the epigenetic signature of gametes (i.e., altered global DNA methylation or aberrant DNA methylation status at specific gene loci, changes in chromatin architecture and the deregulated production of non-coding RNA) emerged with adverse effect on fertilization and early embryo development and possible trans-generational effects and disease load susceptibility in the offspring [4,40, and references therein].

As for instances the exposure to the plasticizer BPA centrally and locally affects spermatogenesis, modulates steroid biosynthesis, induces germ and Sertoli cells apoptosis, interferes in the first round of spermatogenesis, impairs the formation of the blood testis barrier and affects the expression profiles on non-coding RNA and sperm quality [4,52,53,54]. However, different outcomes on male reproduction have been reported in relationship to exposure route, doses, duration and life stage. Studies in humans compared urinary BPA levels to semen parameters, providing evidence of possible BPA association with poorer semen quality [55,56,57]. For example, Pollards and coworkers [56] revealed higher exposure to BPA in association with abnormal sperm tail morphology in a cohort of 161 men aged 18–40 without known subfertility. Omran and coworkers [57] reported a negative association of urinary BPA levels with antioxidant levels and semen quality in terms of motility, morphology and concentration and a positive correlation with DNA damage, and seminal–plasma lipid peroxidation. Lastly, a possible correlation between BPA/phatales metabolites in urine and sperm parameters was investigated, suggesting a greater exposure to EDCs in hypofertile subjects compared to the general population [58]. From here, the need of further studies in the field.

Drug abuse also affects male reproduction. Δ^9^-tetrahydrocannabinol (Δ^9^-THC), the major psychoactive constituent of marijuana plant *Cannabis sativa,* centrally and locally interferes in the endogenous endocannabinoid system—a well-known modulator of spermatogenesis and sperm quality [59,60,61,62]—impairing spermatogenesis and leading to poor semen quality in animal models and humans [63]. Interestingly, recent studies revealed that Δ^9^-THC changes the methylation status at specific gene loci within spermatozoa DNA [40]. As a consequence, in case of fertilization, deregulated epigenetic signature may be transferred from spermatozoa to embryo pointing out the risk of epigenetic impairment of offspring health.

A recent blinded cross-sectional study on a large cohort of 11,706 men, revealed that using an age of 40 years as cut-off value, a significant decrease in routine semen parameters (i.e., volume, count, motility, (progressive motility, total and normal-motile sperm count [TM and NM respectively]), vitality, round cells, and hypo-osmotic swelling (HOS) test) and kinematics (i.e., straight-line velocity (VSL), curvilinear velocity (VCL), average path velocity (VAP), amplitude of lateral head displacement (ALH), mean angular displacement (MAD), and beat cross frequency (BCF)) were age-correlated. In the same cohort, a positive correlation between aging and the percentage of peroxidase-positive cells was found [64]. Similarly, the comparison of sperm quality in healthy subjects over forty with a healthy lifestyle and subjects over forty and exposed to known fertility-compromising factors like genital infections, anabolic steroid consumption, smoking, or exposure to toxics among the others revealed that unhealthy conditions like obesity exerted a significant additional deleterious effect on the semen quality of elderly patients while cigarette smoking and alcohol consumption had only moderate effects. Thus, both aging and unhealthy conditions may contribute the deterioration of sperm quality along the lifespan [64].

The mechanisms decreasing fertility rate during the lifespan are still poorly understood. Apart from a physiological decline in the activity of the reproductive axis, during the aging process, morphological and functional alternations affect the testis, and semen quality declines with changes in sperm morphology and concentration, and defects in the acquisition of sperm motility [43]. At the molecular level, sperm DNA damage, alterations in chromatin architecture mainly due to defective protamination, occur. In parallel, the deregulation of epigenetic marks (i.e., non-coding RNA profile) in both spermatozoa and seminal plasma may affect the subsequent embryonic development and offspring health [65,66]. Several studies have analyzed sperm quality in elderly subjects in physiological or clinical conditions and in relationship to lifestyle. As reported in a recent study from Paoli and co-workers [66], physiologically, semen volume, progressive motility and the number of progressively motile sperm significantly declines in elderly subjects compared to younger subjects. In parallel DNA fragmentation increases, the expression rate of protamines (PRM1 and 2), but not those of transition proteins (TNP1 and TNP2), declines and 67 microRNAs related to pro-inflammatory status, mitochondrial functions, NADPH oxidase complex activity, and also spermatogenesis progression and stem cell exhaustion resulted to be differently expressed in the seminal plasma of elderly subjects compared to younger subjects. Therefore, the relevance of this study is to point out the age-dependent impairment of the suitable microenvironment for the molecular homeostasis of sperm cells and the occurrence of genomic fragility during the physiological aging process. As a consequence, during the aging process, sperm cells become more susceptible to accumulating damage from endogenous or exogenous factors and the fertility rate declines, as a consequence [66].

Diet is also associated to gametogenesis and semen quality in animal models and humans, with centrally mediated and direct mechanisms that primarily affect the suitable hormonal milieu capable of sustaining spermatogenesis [41,67]. The high intake of sweet drinks and snack, processed or red meat, trans and ω6-polyunsaturated fatty acids, low consumption of fish, fruit and vegetable, a low intake of fibers vitamins and minerals negatively affects health and fertility [45]. Obesity and metabolic disorders, hyper caloric or high fat diets are risk factors for male infertility and several trials investigated the possible relation between nutrition and male fertility [68]. In this respect, dietary intervention may help in the preservation of sperm quality along life span and may contribute to the treatment of reproductive dysfunctions. Just to give a few examples, melatonin has been reported to ameliorate the Δ^9^-THC-dependent reduction in hyper-activated sperm hyper motility in capacitated spermatozoa [69], whereas vitamin C ameliorates spermatozoa motility and kinematics in vitro [70]. Resveratrol, a phytochemical found in wine, grapes, rhubarb, blueberries, and peanuts, is able to counteract in vitro the detrimental effects of the polycyclic aromatic hydrocarbon benzo-α-pyrene on human sperm motility, chromatin organization, lipid peroxidation, and the production of mitochondrial superoxide anion [71]. However, exogenous antioxidant supplementation may improve sperm quality along lifespan by reducing oxidative stress, but the outcomes on pregnancy and embryo health are far from being elucidated. Furthermore, caution is necessary to preserve sperm function from the excessive or unnecessary use of antioxidants which may be also capable to inhibit essential and beneficial ROS activities in male reproductive biology.

## 4. Upcoming Evidence on the In Vivo Effects of MPs on Spermatogenesis and SPZ Quality in Mammalian Animal Models

The effects of MPs were studied in the physiology of the testis, revealing the onset of inflammatory states and aberrant production of SPZ.

Hou and coworker [72] orally administered 5µm polystyrene(PS)-MPs in drinking water for 35 days to young male mice (1–5 weeks old animals, doses 100 µg/L, 1000 µg/L, and 10 mg/L in drinking water; for each doses, an estimated average daily dose was 0.6–0.7 μg/day, 6–7 μg/day, and 60–70 μg/day, respectively). After microplastic exposure, the ratio of live sperm in the epididymis to the total number of sperm was significantly lower in exposed animals, with the formation of morphologically aberrant SPZ (i.e., two-tailed, hookless, or swollen neck deformities). In parallel, the morphological analysis of the germinal epithelium within the testis revealed cell damage, a reduced number of spermatids, detached cells from the germinal epithelium, pyknosis and nucleus rupture. Increased expression levels of genes involved in the inflammatory responses (i.e., NF-κBp65 and p-NF-κBp65, Interleukin-1 β (IL-1β), IL-6 and tumor necrosis factor (TNFα)), decreased levels of critical transcriptional factors in the antioxidant defense system and related downstream target (i.e., Nrf2 and HO-1 protein) and increased Bax to Bcl2 ratio and apoptosis were observed [72].

In a further study, following 28 days exposure by oral gavage (100 μL PS-MPs (10 mg/mL)) PS-MPs (0.5, 4, 10 μm) bioaccumulate in mice testis altering spermatogenesis progression, SPZ morphology, testosterone biosynthesis and body weights and inducing inflammatory response [73]; interestingly, in vitro MPs enter into germ cells, Sertoli and Leydig cells [73].

Possible dietary intervention may be useful to protect testis physiology against MPs. In a recent study in mouse, PS-MPs administration by oral gavage (0.01–1 mg/d) resulted in a significant decrease in the number and motility of sperm, an increase in sperm deformity rate and a decrease in testosterone content through the production of ROS and the activation of c-Jun N-terminal kinase (JNK) and p38 mitogen-activated protein kinases(p38MAPK) [74]. Interestingly, SPZ metabolism, evaluated in terms of succinate dehydrogenase (SDH) and lactate dehydrogenase (LDH) activity, was also impaired, but the administration of N-acetylcysteine (NAC) scavenging ROS caused a partial rescue of sperm physiology and improved testosterone biosynthesis [74]. Therefore, antioxidants can mitigate the effects of EDCs and MPs on fertility and metabolism [74].

An intriguing issue concerns the possible additive effects of EDCs at doses considered “safe” for health. Data are still limited, but current hypotheses suggest that MPs may act as “sponges” and transport vectors for hazardous chemicals like heavy metals or pollutants. This way, due to their hydrophobicity and relatively large surface area, MPs may concentrate the absorbed environmental pollutants and carry them into the environment and living organisms, at concentrations that are many orders of magnitude higher than those normally detected in the environment. As a consequence, the microplastic absorbed chemicals may exhibit more toxicity than pure chemicals alone [75,76]. For example, MPs can transport phthalate esters and the combined toxicity of phthalate esters-contaminated MPs have been recently investigated in mouse tissues after 30-day-long exposure [75]. Compared with the exposure to phthalate esters or MPs alone, co-exposure to microplastic–phthalate esters causes higher reproductive toxicity and heavier alterations in spermatogenesis and SPZ physiology, with a higher rate of oxidative stress, and changes into the transcriptome. Interestingly, Deng and co-workers [75] observed a significant accumulation of phthalate esters and MPs at the micro-range in the liver and gut, whereas MPs at the nano range only accumulated in Sertoli cells. Therefore, the potential additive reproductive toxicity due to the co-exposure of MPs and environmental pollutants requires further investigation. However, apart from dose, the microplastic polymer type, size, surface chemistry, and hydrophobicity may also play important roles in their sorption processes of organic pollutants, with saline concentration and pH capable of differently modulating the sorption behavior of contaminants on MPs [76].

The aforementioned observations suggest that MPs may have different effects on reproductive health. In the first place, they can set the conditions for the development of inflammatory states and oxidative stress damage, thus compromising spermatogenesis and SPZ quality; in the second place, they can make living organisms more sensitive to the large plethora of environmental factors that affect male fertility during the lifespan, intensifying and aggravating the toxic effects of EDCs on reproduction. Lastly, the possible generation of nanoplastics (i.e., 100 nm particles of plastic) from polyethylene MPs digestion has been investigated in soil ecosystems using the earthworm *Eisenia andrei* as an experimental model. This study confirmed the possible introduction of nanoplastics into soils through cast excretion, but also revealed impaired coelomocyte viability, defective spermatogenesis and significant damage in the male, but not female, reproductive organs of exposed worms [77]. In a hypothetical scenario in terms of doses (1, 3, 6 and 10 mg/kg–day) and exposure time, nanoplastics of 38.92 nm diameter have been discovered to negatively affect the endocrine control of the reproductive axis in rats, with the occurrence of DNA damage in SPZ which display impaired morphology and viability [78]. Therefore, nanoplastics may represent an additional unraveled threat for fertility.

Taken together, although studies in mammals are still limited, preliminary observations point out a possible risk of MPs for male fertility.

## 5. Conclusions

Evidence regarding MPs toxicity and epidemiology is emerging. Data are still preliminary but suggest that ingested MPs bio-accumulate in mammalian tissue, including the testis, with outcomes on semen quality in rodents, as a consequence of inflammatory state and oxidative stress damage. Effects depend on the size and molecular structure of MPs, thus attention must be given to evaluating their risks to humans and the environment. Furthermore, the morphological features of MPs can make them an ideal vehicle for additional environmental pollutants with EDC activity, and these additive effects are still poorly understood. MPs have been detected in human stool, cancer specimens and placenta, highlighting the possible risk for disease load, successful pregnancy and in utero transmission to the offspring. Furthermore, the high susceptibility of SPZ to ROS along the lifespan, the MPs-dependent production of low-quality SPZ and the metabolic disorders detected in the F1/F2 offspring of MPs exposed dams point out that in rodents, at least, possible parental epigenetic transmission of deregulated epigenetic signature may occur. Spermatogenesis is highly sensitive to environmental pollutants and sperm quality is affected by MPs exposure, at least in animal models. In this respect, the possible reproductive health risks of MPs should not be ignored.

## Figures and Tables

**Figure 1 ijerph-18-02392-f001:**
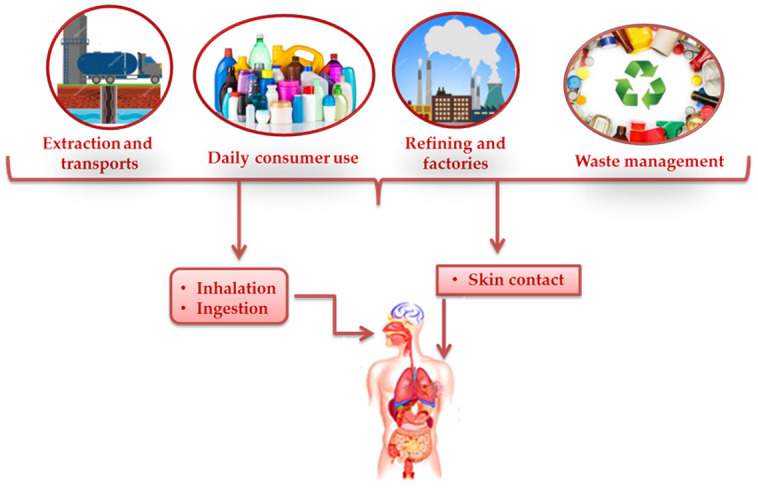
Plastic and human health. Humans are exposed to numerous kinds of toxic chemicals and microplastics through ingestion, inhalation, and direct skin contact, all along the plastic life cycle.

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
