# Peer review of "Microplastics: A Threat for Male Fertility"

_ijerph, 2021, doi:10.3390/ijerph18052392_

Round 1

Reviewer 1 Report

The review “Microplastics: a threat for male fertility” by D’Angelo and Meccariello reports the current knowledge about the impact of microplastics on health, focusing in particular on the effects on male fertility in mammals.

The Authors have indeed long experience on the regulation of male reproduction and possible interference due to exposure to different chemicals.

This review is well written and well subdivided in paragraphs; just check and fix paragraph numbering (3-5).

The review is interesting, since plastic and microplastic pollution is an emerging issue worldwide. The studies of the effects of MPs and their possible associated endocrine disrupting chemicals are still few; nevertheless, this topic is worth to be more studied and this review could therefore serve as an incentive for further research.

Author Response

Thank you very much for your valuable comments. We modified paragraphs numbering.

Reviewer 2 Report

D’Angelo and Meccariello nicely present an overview of the current literature on microplastics present in the environment and their ecotoxicological and health risk in mammals, focusing on male fertility. The following comments are merely provided to improve the quality of the manuscript.

Major comments:

The introduction would benefit from a paragraph explaining why the authors choose to investigate the effect of microplastics on male fertility, as one of the possible negative health effects of MPs. This is not clear from the currently presented information in the introduction.

The authors provide an overview of current literature on the topic. The manuscript would therefore benefit from a methods section describing the search process in finding the presented literature in order to show that all literature published to date is presented.  

Line 73- 75: Do the authors mean 25,000 microns or does this indicate the number of particles, which is suggested in line 75 where they mention that 75,000 tiny microplastic particles are ingested.

Line 77: The “They” that the authors refer to in line 77, do they mean MPs or has this something to do with the contaminants? Please revise

Line 125- 126: Remove “n=” before describing the number of samples

The paragraph ”Sperm quality along life-span” starting on line 134 needs to be changed to section 3 (instead of 2) and revise following paragraph numbering as well.

Can the authors investigate the paragraph spanning from line 180 to 186 as the discussed studies show contradicting results? It looks like some studies even found some positive effects of BPA/ EDC on sperm/ semen quality.

Line 193: what do the authors mean by “for recent review” within the reference parentheses?

Line 302: what do the authors mean by “in such a treatment”

The paragraph on “sperm quality along life-span” misses context and therefore seems long. The article of about the effects of MPs on male fertility. It is known that sperm quality decreases over the life span so in what context do MPs have an (additional) effect on this?

To date, only in vivo studies have been published on the effects of MPs on spermatogenesis according to the authors. The conclusions seem therefore too strong given the provided evidence.

Grammatical errors:

The manuscript should be checked for additional spelling and grammatical errors. Below is a list of errors that need to be corrected:

  • Line 31: Missing comma’s
  • Line 33: “Bisphelol” instead of Bisphenol
  • Line 34: “synthesisof” instead of synthesis of
  • Line 38: remove “an” before ecotoxicological
  • Line 94: replace “ecosistems” by ecosystems
  • Line 119: a word seems to be missing after “as well as”
  • Line 123: revise “occurinmg”
  • Line 125: revise “Althought”
  • Line 127: revise “adnf”
  • Line 127: replace “it suggests” with suggesting
  • Line 130: replace “deserve” with deserves
  • Line 130: replace “heath” with health
  • Line 141: replace “in” with is
  • Line 150: remove “the” in front of others
  • Line 151: replace infertility with infertile
  • Line 152: revise “a high energy level” or “high energy levels”
  • Line 163- 164: please revise “or abuse substances interfere in”
  • Line 175: change “has into have
  • Line 176: change “human” into humans
  • Line 179: replace “association” with associated
  • Line 187: replace “affect” with affects
  • Line 188: revise “psicoactive”
  • Line 196: does it need to be “blinded”
  • Line 205: move “over forty” after the words “subjects
  • Line 220: revise “inelderly”
  • Line 224 and line 229: do the authors mean “in comparison with” instead of “with respect to”?
  • Line 237: revise “mielieu”
  • Line 244: replace “make” with give
  • Line 248: revise “andpeanuts”
  • Line 280: replace “mouse” by mice
  • Line 290: do the authors mean limited instead of “poor”?
  • Line 300: revise “esposure”
  • Line 302: revise “trascriptome”
  • Line 311: replace “study” by studies
  • Line 318: replace “must be given at” to “must be given to”
  • Line 321: please revise “human stolen”

Author Response

D’Angelo and Meccariello nicely present an overview of the current literature on microplastics present in the environment and their ecotoxicological and health risk in mammals, focusing on male fertility. The following comments are merely provided to improve the quality of the manuscript.

Response to the comments of Reviewer #2

Thank you very much for your valuable comments. We have revised the manuscript in accordance with your suggestions. The comments and our responses are numbered and presented as follows:

Major comments:

  1. The introduction would benefit from a paragraph explaining why the authors choose to investigate the effect of microplastics on male fertility, as one of the possible negative health effects of MPs. This is not clear from the currently presented information in the introduction.

We thank the reviewer for this comment. We have included the following statements into the introduction:

In recent years, the growing rate of infertility has displaced attention to gametogenesis and gamete quality. Once considered a “woman’s trouble”, infertility is on the rise in males and semen quality has declined over the last decades [11]. In about 40% of men with impaired spermatogenesis, the aetiology remains unknown after a complete diagnostic work-up [12]. SPZ is not only the carrier of a haploid nucleus into egg cells, but emerging evidence has revealed that SPZ contribute to early embryo development and offspring health with an epigenetic signature highly sentitive both to environmental factors, including EDCs, and paternal life style [4]. Decreased sperm quality has been reported in subjects exposed to environmental pollutants [13, 14, for review], suggesting that the evaluation of sperm quality may represent a biomarker for general health [13, 14].”.

  1. The authors provide an overview of current literature on the topic. The manuscript would therefore benefit from a methods section describing the search process in finding the presented literature in order to show that all literature published to date is presented.

In literature only few studies are available in mammals, but their number is increasing. At the submission of this manuscript pubmed search (entrez microplastics and testis, microplastics and sperm, microplastics and male reproductive toxicity in mammals) revealed 7 peer-reviewed manuscripts only, all published between 2019-2021: 4 research articles analysed the in vivo effects of MPs in mice (all cited in the previous version of the manuscript); 1 research article was on nanoplastics in testis (not cited in the previous version of the manuscript), 2 research articles were on maternal/parental exposure (only the manuscript on maternal exposure was cited in the previous version of the manuscript). During the revision on this manuscript, in Feb 2021 two manuscripts on the effects of MPs on rat ovary appeared in pubmed; 1 manuscript reported that earthworm has the ability to digest MPs into nanoplastics.

In the current version of the manuscript we have updated references at early February 2021, with the discussion of manuscripts reporting MPs effects in ovary (REF. 31, 32 both dated February 2021), the effects of MPs parental exposure on immune system and offspring size (REF 33 dated 2020), the digestion of MPs in nanoplastics by earthworm (REF 77 dated Feb 2021) and the effects of nanoplastics in mammals testis (ref 78 dated 2020).

In introduction section we have included the following sentence: “However, in the past three years, preliminary assessments of the effects of MPs exposure in mammalian reproduction have emerged with the publication of peer-review articles that revealed the effects on spermatogenesis and sperm quality in exposed animal models and the indirect effects on the offspring occurring via gestational exposure.”

  1. Line 73- 75: Do the authors mean 25,000 microns or does this indicate the number of particles, which is suggested in line 75 where they mention that 75,000 tiny microplastic particles are ingested.

The number 25,000 indicates the number of particles, with a size of microns.

We replaced by: “They observed that about 25,000 micron-sized microplastic particles are released into 100 ml of hot liquid (85 to 90 °C)”.

  1. Line 77: The “They” that the authors refer to in line 77, do they mean MPs or has this something to do with the contaminants? Please revise.

We replaced “They” by “Dangerous MPs”.

  1. Line 125- 126: Remove “n=” before describing the number of samples.

We corrected.

  1. The paragraph ”Sperm quality along life-span” starting on line 134 needs to be changed to section 3 (instead of 2) and revise following paragraph numbering as well.

We corrected.

  1. Can the authors investigate the paragraph spanning from line 180 to 186 as the discussed studies show contradicting results? It looks like some studies even found some positive effects of BPA/ EDC on sperm/ semen quality.

We have carefully revised this part of the manuscript 5trying to avoid any misunderstandings and adding clarity as requested. Text has been changes as follows: “Lastly, a possible correlation between BPA/phatales metabolites in urine and sperm parameters was investigated suggesting a greater exposure to EDCs in hypofertile subjects compared to the general population”

  1. Line 193: what do the authors mean by “for recent review” within the reference parentheses?

We have deleted “for recent review”

  1. Line 302: what do the authors mean by “in such a treatment”

We have deleted “in such a treatment”

  1. The paragraph on “sperm quality along life-span” misses context and therefore seems long. The article of about the effects of MPs on male fertility. It is known that sperm quality decreases over the life span so in what context do MPs have an (additional) effect on this?

This point has been explained as follows in the manuscript:

“The aforementioned observations suggest that MPs may have different effects on reproductive health. In the first place, they can set the conditions for the development of inflammatory states and oxydative stress damage, thus compromising spermatogenesis and SPZ quality; in the second place, they can make living organisms more sensitive to the large plethora of environmental factors that affect male fertility during the life-span, intensifying and aggravating the toxic effects of EDCs on reproduction. Lastly, the possible generation of nanoplastics (i.e. 100 nm particles of plastic) from polyethylene MPs digestion has been investigated in soil ecosystems using the earthworm Eisenia andrei as experimental model. This study confirmed the possible introduction of nanoplastics into soils through cast excretion, but also revealed impaired coelomocyte viability, defective spermatogenesis and significant damage in the male, but not female, reproductive organs of exposed worms [77]. In a hypothetical scenario in terms of doses (1, 3, 6 and 10 mg/kg-day) and exposure time, nanoplastics of 38.92 nm diameter have beed discovered to negatively affect the endocrine control of reproductive axis in rats, with the occurrence of DNA damage in SPZ which display impaired morphology and viability [78]. Therefore, nanoplastics may represent an additional unravelled threat for fertility.”

  1. To date, only in vivo studies have been published on the effects of MPs on spermatogenesis according to the authors. The conclusions seem therefore too strong given the provided evidence.

Accordingly to reviewer’s suggestion we have changes the last part of the conclusion as follows: Furthermore, the high susceptibility of SPZ to ROS along the life-span, the MPs dependent production of low quality SPZ and the metabolic disorders detected in the F1/F2 offspring of MPs exposed dams point out that in rodents, at least, possible parental epigenetic transmission of deregulated epigenetic signature may occur. Spermatogenesis is highly sensitive to environmental pollutants and sperm quality is affected by MPs exposure, at least in animal models. In this respect, the possible reproductive health risks of MPs should not be ignored.

Grammatical errors:

The manuscript has been checked.

We corrected the indicated grammatical errors 

Reviewer 3 Report

I believe thar the review is well articulated and that it synthesizes very well the knowledge on an emerging problem that deserves to be more widely known

Author Response

We thank the reviewer for her/his valuable comments.

Round 2

Reviewer 2 Report

Dear authors,

Thank you for the revised version of your manuscript following the suggestions. Can you please revise the following below:

  • Please explain the abbreviation "SPZ" at first use (so on page 2 line 46 instead of page 5, line 158)

Author Response

We have carried out the suggested change.